# The Co-Expression of Programmed Death-Ligand 1 (PD-L1) in Untreated EGFR-Mutated Metastatic Lung Adenocarcinoma

**DOI:** 10.3390/biomedicines8020036

**Published:** 2020-02-19

**Authors:** Ping-Chih Hsu, Chih-Wei Wang, Scott Chih-Hsi Kuo, Shu-Min Lin, Yu-Lun Lo, Allen Chung-Cheng Huang, Li-Chung Chiu, Cheng-Ta Yang

**Affiliations:** 1Division of Thoracic Medicine, Department of Internal Medicine, Chang Gung Memorial Hospital at Linkou, Chang Gung University College of Medicine, Taoyuan City 33305, Taiwan; 8902049@gmail.com (P.-C.H.); chihhsikuo@gmail.com (S.C.-H.K.); smlin100@gmail.com (S.-M.L.); loyulun@hotmail.com (Y.-L.L.); mr0818@cgmh.org.tw (A.C.-C.H.); cremaster54@yahoo.com.tw (L.-C.C.); 2Department of Pathology, Chang Gung Memorial Hospital, Linkou, Chang Gung University College of Medicine, Taoyuan City 3305, Taiwan; weger@cgmh.org.tw; 3Department of Respiratory Therapy, College of Medicine, Chang Gung University, Taoyuan City 33302, Taiwan

**Keywords:** epidermal growth factor receptor (EGFR), tyrosine kinase inhibitor (TKI), programmed death-ligand 1 (PD-L1), lung adenocarcinoma, metastasis

## Abstract

Epidermal growth factor receptor (EGFR)-tyrosine kinase inhibitor (TKI) is the standard first-line therapy for metastatic lung adenocarcinoma harboring sensitive EGFR mutations. Tumor surface programmed death-ligand 1 (PD-L1) is expressed in some metastatic EGFR-mutated lung adenocarcinoma, but its impact on the efficacy of EGFR-TKIs is unclear. We retrospectively investigated 117 untreated metastatic lung EGFR mutated adenocarcinoma patients with a PD-L1 immunohistochemistry test. The PD-L1 expression level was classified by tumor proportion scores (TPS). Forty-five patients had negative expression (TPS < 1%), 45 had a weak expression (TPS 1–49%), and 27 had a strong expression (≥50%). All patients recruited in this study received EGFR-TKIs as a first-line therapy. No significant differences were observed for objective response rates (68.9% versus 62.2% versus 73.1%, *p* = 0.807) and median time to treatment failure (TTF) (12.17 versus 13.17 versus 11.0 months, *p* = 0.443) of first-line EGFR-TKIS among the three groups of patients (negative versus weak versus strong). The median overall survival was 21.27 versus 20.63 versus 19.43 months among the three groups of patients (*p* = 0.77). Our results demonstrated that PD-L1 did not affect the efficacy of first-line EGFR-TKIs in metastatic EGFR mutated lung adenocarcinoma. Thus, EGFR-TKIs are suggested as the preferred clinical therapy for these patients, despite their PD-L1 levels.

## 1. Introduction

Advanced and surgically unresectable lung cancer is the leading cause of cancer-related mortality in both men and women worldwide [1]. According to the classification of histopathology, non-small-cell lung cancer (NSCLC) accounts for 85% of cases, and adenocarcinoma is the most common histological type of NSCLC (~50%) [2,3]. Active mutation in the epidermal growth factor receptor (EGFR) kinase domain has been well studied and is known to be the main oncogenic-driven mutation in lung adenocarcinoma [4,5]. The frequency of EGFR-driven mutations in lung adenocarcinoma rates ranges from 40% to 55% in East Asians [5,6]. L858R and exon 19 deletion are the two types of the most frequent EGFR mutations and account for 90% of all EGFR mutations [5,6,7]. EGFR-tyrosine kinase inhibitors (TKIs), including gefitinib, erlotinib, and afatinib, have shown promising efficacy (60–80% response rate and 9–13 months of progression-free survival) in treating advanced lung adenocarcinoma harboring L858R or exon 19 deletion mutations in clinical trials [7,8,9]. Therefore, EGFR-TKIs are used as a standard first-line therapy for advanced lung adenocarcinoma patients with sensitive EGFR mutations (L858R or exon 19 deletion) [8,9,10].

Programmed death-ligand 1 (PD-L1; also known as B7-H1 or CD274) is a ligand of programmed cell death protein 1 (PD-1; also known as CD279), and the PD-L1/PD-1 pathway is known as an immune checkpoint. In normal human tissue cells, the binding of PD-L1 and PD-1 promotes T-cell tolerance by downregulating CD8+ T-cell survival and effector function [11]. PD-L1 expression has been reported to appear in human NSCLC, and NSCLC cells can escape the anti-tumor immune response of their host by engaging the immune checkpoint PD-L1/PD-1 axis [12,13,14]. In response, anti -PD-L1/PD-1 immune checkpoint inhibitors have been developed as new therapies for advanced NSCLC [13,14,15,16,17,18]. PD-L1 expression on NSCLC tumor surfaces has been used as a predictive biomarker for anti-PD-L1/PD-1 immunotherapy, and PD-L1-positive NSCLC is correlated with a higher response to anti-PD-L1/PD-1 immunotherapy [15,19]. PD-L1 has been reported to be expressed in some EGFR mutant NSCLC patients [20], but the impact of PD-L1 expression on EGFR mutant NSCLC is not clear. Here, we sought to investigate whether PD-L1 expression affects the efficacy of EGFR-TKIs and the clinical outcome in untreated metastatic EGFR-mutated lung adenocarcinoma patients.

## 2. Materials and Methods

### 2.1. Patients

From July 2016 to June 2017, 253 newly diagnosed stage IV lung adenocarcinoma and treatment-naïve patients registered in the Cancer Center Registry System of Chang Gung Memorial Hospital (CGMH) in Linkou, a university-affiliated hospital in Taiwan, were screened. A total of 117 patients were retrieved and analyzed in this study. The inclusion criteria were (1) patients diagnosed with lung adenocarcinoma via tissue samples taken from a core-needle biopsy, pleural biopsy, transbronchial lung biopsy, or surgical specimens; (2) patients with common EGFR mutations that are sensitive to EGFR-TKI (L858R and exon 19 deletion); (3) patients having received gefitinib, erlotinib, or afatinib as their first-line treatment for lung cancer, and (4) a PD-L1 expression test that was done by immunohistochemistry staining at initial diagnosis. Patients were excluded for the following reasons: (1) no EGFR mutation, (2) uncommon or T790M mutations, and (3) EGFR mutations. PD-L1 tests were done on different tissue samples. Figure 1 summarizes the inclusion criteria for retrieving patients in this study.

Clinical information, including complete medical history, physical examination, and treatments for lung cancer, were recorded in the Cancer Center Registry System at CGMH. Baseline images, including chest X-ray (CXR), computed tomography (CT) of the chest and abdomen, brain magnetic resonance imaging (MRI) of the head, and fluorodeoxyglucose positron-emission tomography (FDG-PET), were performed at diagnosis to determine the stage of disease. All patients received at least a whole-body CT during their EGFR-TKIs therapies to evaluate their treatment efficacy, and additional images, including CXR, PET scan, and MRI, were performed depending on the clinical physician’s judgement. All images were interpreted by independent radiologists at CGMH.

Retrieval and analysis of information in the database was approved by the Institutional Review Board (IRB) (No.201901736B0) of CGMH, and the requirement for obtaining personal informed consent was waived by the IRB.

### 2.2. EGFR Mutation and PD-L1 Expression Tests

Both EGFR mutation and PD-L1 were assessed in formalin-fixed tumor samples in the Central Molecular Lab of the Department of Pathology at CGMH, a College of American Pathologists (CAP)-accredited laboratory. The EGFR mutations were detected by using Amplified Refractory Mutation System (ARMS)-Scorpion methods [21]. PD-L1 expression was assayed by using an immunohistochemistry (IHC) 22C3 pharmDx assay (Dako North America) [22]. The PD-L1 IHC staining of tumor samples was interpreted by experienced pathologists and assessed by its tumor proportion score (TPS). PD-L1 expression was reported by using a three cut-point system: negative, TPS < 1%; weak, TPS 1–49%; and strong, TPS ≥ 50%.

### 2.3. Evaluation of the Efficacy of EGFR-TKIs Therapy

Treatment responses were defined as progressive disease (PD), stable disease (SD), or partial response (PR) according to the criteria in the “Criteria for Solid Tumors group” from “Response Evaluation Criteria in Solid Tumors (RECIST)” Version 1.1. The time-to-treatment failure (TTF) was defined as the period from the first date of the EGFR-TKI therapy until the last date of the regimen. The TTF values for subjects who were still receiving EGFR-TKIs at the time of the final follow-up date were registered as censored at the date of the last tumor assessment. Overall survival (OS) was defined as the period from the date of diagnosis until the date of death. For patients who were alive at the time of the final follow-up date, survival was censored at the date of the last visit of follow-up.

### 2.4. Statistical Analysis

Data for age and PD-L1 expression level are presented as the mean ± standard deviation. Column statistics were conducted for continuous variables to compare the means between the three groups. Frequency distributions between the three groups were tested using a Pearson’s chi-squared test. Survival rates were calculated by using the Kaplan–Meier method, and comparison of survival curves was based on a log-rank test. All *p* values were two-sided, and *p* < 0.05 was considered statistically significant. GraphPad Prism (Version 5.0; GraphPad Software, San Diego, CA, USA) was used for the statistical analyses.

## 3. Results

### 3.1. Patient Baseline Characteristics of This Study

Between June 2016 and October 2018, 117 stage IV lung EGFR mutant adenocarcinoma patients were retrieved and analyzed; Figure 1 summarizes the way that the patients were recruited in this study. The patient characteristics are shown in Table 1. Among the 117 patients, 45 had negative PD-L1 expression, 45 had weak PD-L1 expression (TPS 1–49%), and 27 had strong PD-L1 expression (≥50%). All patients received EGFR-TKIs as their first-line therapy. Seventy-five (64%) patients took afatinib, 27 (23%) took erlotinib, and the other 15 (13%) took gefitinib during first-line EGFR-TKI therapy. Patients were divided into three groups according to their PD-L1 expression levels. Examples of the three groups of patients are shown in Figure 2. There was no statistically significant difference in the baseline characteristics between the three groups of patients (Table 2).

### 3.2. Response Rate of First-Line EGFR-TKIs

Among all 117 patients of this study, 79 had a partial response (PR) to first-line EGFR-TKI treatment, 20 had stable disease (SD), and 18 had progressive disease (PD). The objective response rate and disease control rate were 67.5% and 84.6%, respectively, among all 117 patients (Figure 3A). There was no statistically significant difference in a comparison of the response rates to first-line EGFR-TKI therapy between the three groups of patients divided by PD-L1 expression level (*p* = 0.807) (Figure 3B).

### 3.3. Time to Treatment Failure (TTF) for First-Line EGFR-TKIs and Overall Survival (OS)

For all the patients of this study, the median TTF of the first-line EGFR-TKIs was 12.2 months, and the median OS was 24.1 months (Figure 4A,B). When the patients were divided into three groups by different PD-L1 expression levels, the median TTFs of the first-line EGFR-TKIs were 12.2, 13.2, and 11 months for the negative, weak, and strong groups, respectively (*p* = 0.443) (Figure 4C). The median OS rates were 24.1, 20.6, and 19.4 for the negative, weak, and strong groups (*p* = 0.77) (Figure 4D). There was no statistically significant difference in the TTFs for the first-line EGFR-TKIs or the OS between the three groups.

## 4. Discussion

The results of our retrospective study offer some clinical insights on advanced lung cancer harboring sensitive EGFR mutations. First, we demonstrated that PD-L1 is concurrently expressed on the tumor surface during part of stage IV EGFR mutant lung adenocarcinoma. Second, different PD-L1 expression levels did not affect the efficacy of first-line EGFR-TKIs in treating stage IV EGFR mutant lung adenocarcinoma patients, neither in response nor TTF. In addition, the OS of the patients in our study was not affected by the PD-L1 expression level.

The proportion of strong PD-L1 (TPS ≥ 50%) expression in our study was 24% and higher than that of a previous study (9.9% in EGFR-mutated lung adenocarcinoma) [23]. The different numbers of patients and tissue samples may contribute to that difference. In the study of Yoneshima et al., the total number of EGFR-mutated patients was 71 less than ours (*n* = 117). Furthermore, tissue samples from endobronchial ultrasound-guided transbronchial needle aspiration or effusion cell block were allowed in their study but were not included in our study [23].

The response rate of first-line EGFR-TKIs in this study was 60–70%, which is similar to that of previous clinical trials or real-world data analysis [7,8,9,10]. We evaluated the efficacy of first-line EGFR-TKIs by using TTF, not progression-free survival (PFS), because EGFR-TKI treatment sometimes continues beyond disease progression, which is defined by the RECIST criteria of an image. Previous studies have reported that continuous EGFR-TKI therapy beyond RECIST-defined progression partially benefits EGFR-mutant advanced NSCLC patients clinically [24,25]. The overall TTF of first-line EGFR-TKIs for our study is 12.2 months, which is within the PFS ranges shown in previous studies (ranging from 9 to 13 months) [7,8,9,10].

Two previous studies reported that strong PD-L1 expression contributed to a poor prognosis of advanced EGFR-mutated NSCLC patients receiving EGFR-TKIs [23,26]. The two studies showed that EGFR-mutated NSCLC patients with strong PD-L1 expression had a lower response rate and a shorter PFS than those with negative PD-L1 expression [23,26]. In one of the two studies, ALK rearrangement (*n* = 9) of NSCLC, which was excluded in our study, was included and analyzed, and most patients (50%) took gefitinib (first-generation EGFR-TKI) [23]. The majority of our study patients received afatinib (second-generation EGFR-TKI) (64%) as a first-line therapy. In the other study, four PD-L1-positive NSCLC patients with de novo resistance to EGFR-TKIs were found to have EGFR-TKI-resistant mutations, MET amplification, and K-ras G12D mutations in a genetic profile analysis [26]. This result may suggest that PD-L1-positive NSCLC patients have a lower response rate and PFS because of resistant mutations, not because of PD-L1. This study, however, did not show which EGFR-TKIs were given to their study patients [26]. The numbers of patients with positive PD-L1 expression in the two studies were 35 and 36 [23,26], and fewer patients were included in our study (*n* = 72). Together, the different patient numbers, genetic profiles, and EGFR-TKIs may be the reasons why PD-L1 expression affected the prognosis of EGFR-mutated NSCLC patients in the two previous studies but not in our study.

A recent study investigated whether PD-L1 expression is associated with clinical outcomes following treatment with osimertinib (third-generation EGFR-TKI) or comparator EGFR-TKIs (gefitinib/erlotinib) in the FLAURA trial (phase III, NCT02296125) [27,28]. The authors concluded that the efficacy and clinical benefit of osimertinib were not affected by PD-L1 expression status [28]. In the comparator group of the FLAURA trial, PD-L1 expressers had a 71% response rate and 6.9 month median duration in their response to EGFR-TKIs; this result indicates that positive PD-L1 expression did not affect the efficacy of gefitinib and erlotinib in treating EGFR-mutated NSCLC patients [28]. In previous preclinical studies, the EGFR signaling pathway was reported to regulate PD-L1 expression in EGFR-mutated NSCLC cell lines [29,30,31]. These studies all showed that the inhibition of EGFR by EGFR-TKIs downregulates PD-L1 expression in NSCLC cell lines [29,30,31]. Together, the findings of the FLAURA trial and preclinical studies may support the results of our study: that PD-L1 expression does not affect the efficacy of EGFR-TKIs or survival in untreated stage IV EGFR-mutated lung adenocarcinoma patients.

Our study has some limitations because the patients in our study were all East-Asian, and only exon 19 deletion and L858R EGFR mutations were included. Whether racial groups other than East-Asians and those with other mutations (such as T790M, G719X, L861Q, and S768I) will have the same results as those of our study will require further studies to be explored.

## 5. Conclusions

PD-L1 expression does not affect the efficacy of first-line EGFR-TKIs in untreated stage IV EGFR-mutated lung adenocarcinoma. EGFR-TKIs are thus suggested as first-line therapies for stage IV EGFR-mutated lung adenocarcinoma patients, despite their PD-L1 expression levels in clinical practice.

## Figures and Tables

**Figure 1 biomedicines-08-00036-f001:**
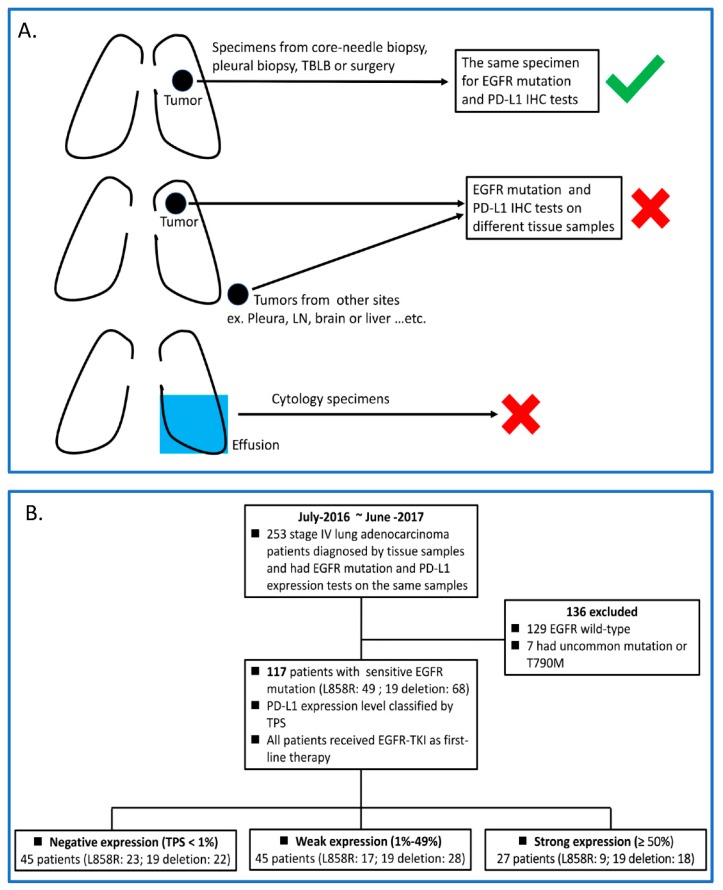
(**A**) Scheme of patient inclusion and exclusion criteria in this study; (**B**) Flow chart of the study. TBLB, transbronchial lung biopsy; EGFR, epidermal growth factor receptor; TKIs, tyrosine kinase inhibitors; PD-L1, programmed death ligand 1; IHC, immunohistochemistry; TPS, tumor proportion scores.

**Figure 2 biomedicines-08-00036-f002:**
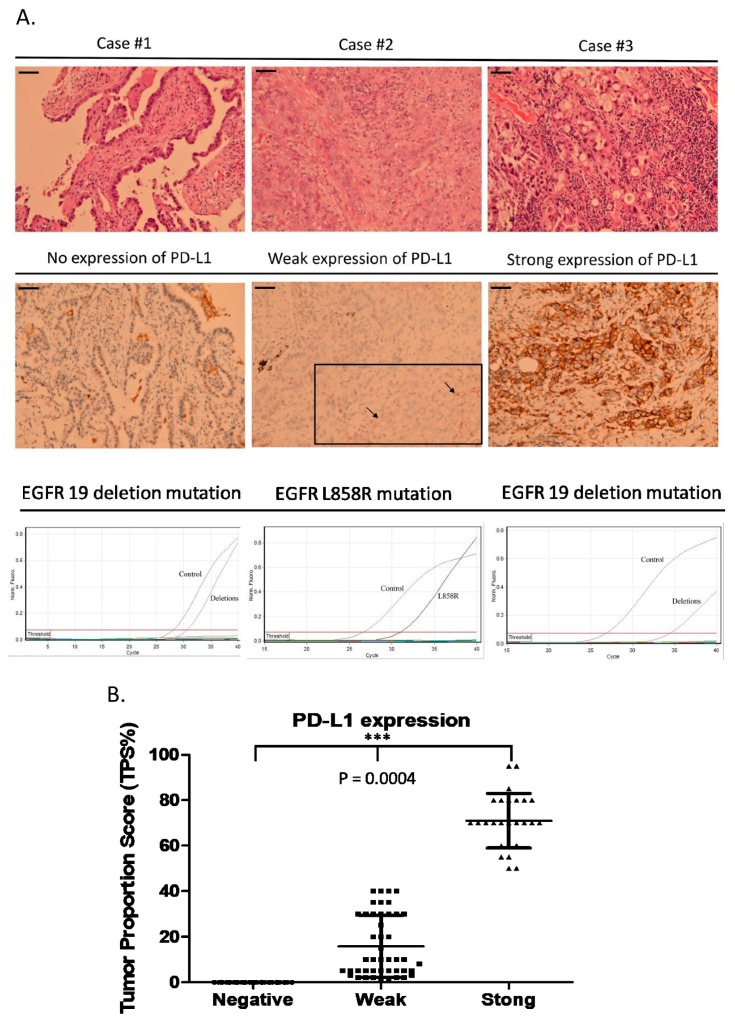
Examples of the three groups of patients with different PD-L1 expression levels. (**A**) Case #1 EGFR exon 19 deletion mutation without PD-L1 expression; case #2 EGFR L858R mutation with weak PD-L1 expression; case #3 EGFR exon 19 deletion with strong PD-L1 expression. (**B**) Comparison of PD-L1 expression levels among the three groups (mean ± SD %) (0 versus 15.8 ± 13.2 versus 70.9 ± 12, *p* = 0.0004). The scale bar = 100 µm.

**Figure 3 biomedicines-08-00036-f003:**
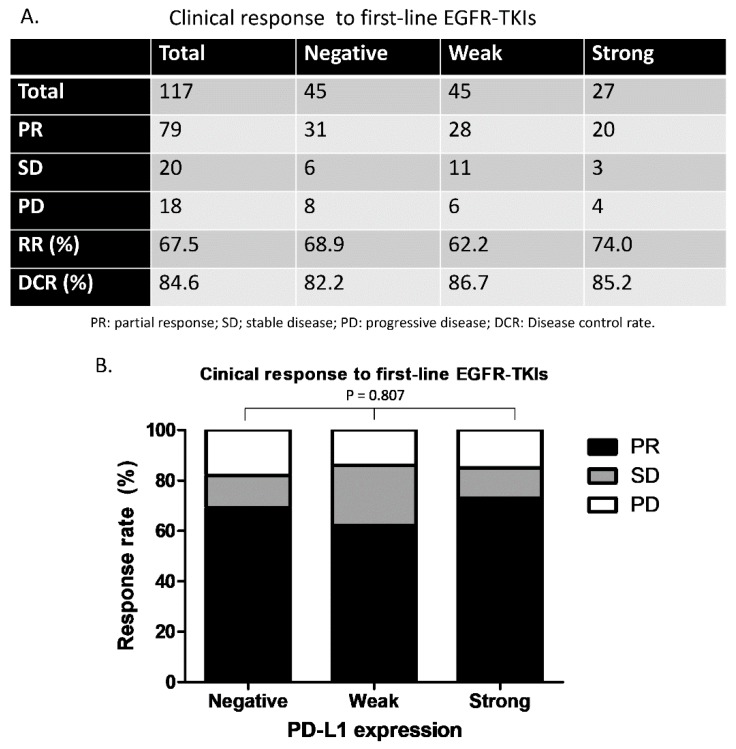
Clinical response to first-line EGFR-TKIs in this study. (**A**) Overall response rate and disease control rate for the three groups of patients. (**B**) Comparison of the response rate between the three groups of patients (*p* = 0.807). PR, partial response; SD, stable disease; PD, progressive disease; DCR, disease control rate.

**Figure 4 biomedicines-08-00036-f004:**
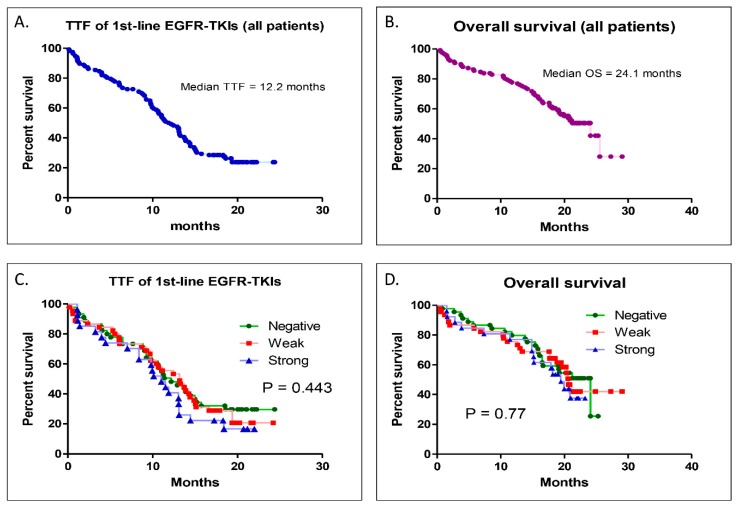
Kaplan–Meier survival analysis of time to treatment failure (TTF) and overall survival (OS). (**A**) TTF for the overall patients of this study. (**B**) OS for all patients of this study. (**C**) Comparison of the TTF between the three different PD-L1 level groups. (**D**) Comparison of the OS between the three different PD-L1 level groups.

**Table 1 biomedicines-08-00036-t001:** Overall patient baseline characteristics.

	No.	(%)
Total	117	
SEX		
male	40	34
female	77	66
AGE (mean ± SD)	64.5 ± 11.2	
ECOG PS		
0–1	86	73.5
≥2	31	26.5
SMOKING STATUS		
Non-smoker	83	71
Former + Current smoker	34	29
Adenocarcinoma	117	100
Stage		
IV	117	100
EGFR Mutation		
L858R	49	42
Exon 19 deletion	68	58
PD-L1 expression (TPS%)		
Negative (TPS < 1%)	45	38
Weak (TPS 1–49%)	45	38
Strong (TPS ≥ 50%)	27	24
First-line EGFR-TKIs		
Afatinib	75	64
Erlotinib	27	23
Gefitinib	15	13

SD, Standard Deviation; ECOG PS, Eastern Cooperative Oncology Group Performance Status; TPS, Tumor Proportion Scores; EGFR, epidermal growth factor receptor; TKIs, tyrosine kinase inhibitors.

**Table 2 biomedicines-08-00036-t002:** Comparison of characteristics between three groups of patients.

	Negative(TPS < 1%)	Weak(TPS 1%–49%)	Strong(TPS ≥ 50%)	*p*-Value
Total	45	45	27	
SEX				0.09
male	11	16	13	
female	34	29	14	
AGE (mean ± SD)	65.2 ± 10.4	63.7 ± 12.3	64.9 ± 11.0	0.93
ECOG PS				
0–1	29	36	21	0.93
≥2	16	9	6	
SMOKING STATUS				
Non-smoker	36	33	14	0.074
Former + Current smoker	9	12	13	
EGFR mutation				0.111
L858R	23	17	9	
19 deletion	22	28	18	
PD-L1 expression (TPS%) (mean ± SD %)	0	15.8 ± 13.2	70.9 ± 12	0.0004
Metastatic sites				
<3	33	36	21	0.613
≥3	12	9	6	
First-line EGFR-TKIs				0.471
Afatinib	26	28	21	
Erlotinib	13	11	3	
Gefitinib	6	6	3

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
