# Peer review of "The Co-Expression of Programmed Death-Ligand 1 (PD-L1) in Untreated EGFR-Mutated Metastatic Lung Adenocarcinoma"

_biomedicines, 2020, doi:10.3390/biomedicines8020036_

Round 1
Reviewer 1 Report
I like to commend the authors for undertaking this study. Although the result did not reveal any cross-interaction between EGFR and PD-1 pathway, it is important to have these negative study in the face of other reported positive study. The interaction between EGFR and immune checkpoint may be independent pathways that may not cross-talk, but we would not have any idea until we have larger data-set. This study provides a small subset of that information in a particular population in Taiwan. This study contributes additional information to the overall understanding of the interaction between these two pathways.
The study is conduct in a clean fashion without significant issues.
I would only suggest that some of the English could be improved to provide more active voice and better clarity.
Otherwise, I would recommend the paper be accepted with grammar revision.
Author Response
Point 1: I would only suggest that some of the English could be improved to provide more active voice and better clarity. 

Response 1: English language editing of revised manuscript was done as suggested.
Reviewer 2 Report
In this article, authors present a retrospective study of lung cancer specimen, concluding that PD-L1 expression does not affect the efficacy of EGFR-TKIs. In my opinion the manuscript is pretty well-written and flowing and could be useful for clinicians.
Author Response
Point 1: In this article, authors present a retrospective study of lung cancer specimen, concluding that PD-L1 expression does not affect the efficacy of EGFR-TKIs. In my opinion the manuscript is pretty well-written and flowing and could be useful for clinicians. 

Response 1: We appreciate the reviewer’s enthusiastic comments about our research article.
This manuscript is a resubmission of an earlier submission. The following is a list of the peer review reports and author responses from that submission.